# One hundred years of zoonoses research in the Horn of Africa: A scoping review

Lisa Cavalerie[1,2]*, Maya Wardeh[1,3], Ophélie Lebrasseur[2,4], Mark Nanyingi[1,5], K. Marie McIntyre[1,6], Mirgissa Kaba[7], Daniel Asrat[8], Robert Christley[1,6], Gina Pinchbeck[1], Matthew Baylis[1,6], Siobhan M. Mor[1,2]

**1** Department of Livestock and One Health, Institute of Infection, Veterinary and Ecological Sciences, University of Liverpool, Brownlow Hill, Liverpool, United Kingdom, **2** International Livestock Research Institute, Addis Ababa, Ethiopia, **3** Department of Mathematical Sciences, University of Liverpool, Peach Street, Liverpool, United Kingdom, **4** Department of Archaeology, Classics and Egyptology; School of Histories, Language and Cultures, University of Liverpool, Liverpool, United Kingdom, **5** Department of Epidemiology and Public Health, School of Public Health, University of Nairobi, Nairobi, Kenya, **6** Health Protection Research Unit in Emerging and Zoonotic Infections, University of Liverpool, Liverpool, United Kingdom, **7** School of Public Health, Addis Ababa University, Addis Ababa, Ethiopia, **8** Department of Microbiology, Immunology and Parasitology, School of Medicine, College of Health Sciences Addis Ababa University, Addis Ababa, Ethiopia

* lisa.cavalerie@liverpool.ac.uk

**Data Availability Statement:** All relevant data are within the manuscript and its Supporting information files.

## Abstract

### Background

One Health is particularly relevant to the Horn of Africa where many people's livelihoods are highly dependent on livestock and their shared environment. In this context, zoonoses may have a dramatic impact on both human and animal health, but also on country economies. This scoping review aimed to characterise and evaluate the nature of zoonotic disease research in the Horn region. Specifically, it addressed the following questions: (i) what specific zoonotic diseases have been prioritised for research, (ii) what data have been reported (human, animal or environment), (iii) what methods have been applied, and (iv) who has been doing the research?

### Methodology/principal findings

We used keyword combinations to search online databases for peer-reviewed papers and theses. Screening and data extraction (disease, country, domain and method) was performed using DistillerSR. A total of 2055 studies focusing on seven countries and over 60 zoonoses were included. Brucellosis attracted the highest attention in terms of research while anthrax, Q fever and leptospirosis have been comparatively under-studied. Research efforts did not always align with zoonoses priorities identified at national levels. Despite zoonoses being a clear target for 'One Health' research, a very limited proportion of studies report data on the three domains of human, animal and environment. Descriptive and observational epidemiological studies were dominant and only a low proportion of publications were multidisciplinary. Finally, we found that a minority of international collaborations were between Global South countries with a high proportion of authors having affiliations from outside the Horn of Africa.

**Funding:** LC, OL, MN, KMM, MK, DA, RC, GP, MB, SMM were supported by the Global Challenges Research Fund (GCRF) One Health Regional Network for the Horn of Africa (HORN) Project, from UK Research and Innovation (UKRI) and Biotechnology and Biological Sciences Research Council (BBSRC) (project number BB/P027954/1). The funders had no role in study design, data collection and analysis, decision to publish, or preparation of the manuscript.

## Conclusions/significance

There is a growing interest in zoonoses research in the Horn of Africa. Recommendations arising from this scoping review include: (i) ensuring zoonoses research aligns with national and global research agendas; (ii) encouraging researchers to adopt a holistic, transdisciplinary One Health approach following high quality reporting standards (COHERE, PRISMA, etc.); and (iii) empowering local researchers supported by regional and international partnerships to engage in zoonoses research.

## Author summary

Zoonoses are diseases that are transmissible between animals and humans. Some emerging or re-emerging zoonoses, like avian influenza, regularly make the headlines in international media. Others, like rabies or echinococcosis, which mainly affect poor communities, attract much less attention, and are considered neglected by the World Health Organisation (WHO). In the Horn of Africa, many people live in close proximity to livestock and depend on them for food and income. Their frequent interaction with animals increases the risk of contracting zoonoses. In our work, we have searched for existing research publications on zoonoses in the Horn of Africa to guide future research on most neglected areas. Based on 2055 publications, we have described which zoonoses have been studied where and using which method. Notably, we found that very limited research followed One Health approaches. That implies that separate focus was given to animals or humans and a single method or discipline was used, while the One Health approach advocates for multidisciplinary and multisectoral collaboration to address complex issues like zoonoses. Finally, we identified that a majority of authors were affiliated with countries from the Global North which hinders relevance, equity and sustainability of Global North-Global South research collaborations.

## Introduction

In the Horn of Africa, the livelihood of millions of people is highly, and in some cases entirely, dependent on livestock. The region is home to 280 million people [1], including some 60 million pastoralists, as well as more than 400 million ruminants—10% of the total ruminant population on the planet—and 220 million monogastrics (chickens and pigs) [2]. Livestock provides about 60% of agricultural Gross Domestic Product (GDP) in the Horn of Africa [3,4] and is a major source of employment and foreign currency [5–7]. Annual exports of live animals from the region and neighbouring countries, to the Middle East in particular, are estimated at close to US$ 1 billion [8]. Wildlife is also very important to the ecosystem and livelihoods of the Horn of Africa as the region has been recognised as one of 36 biodiversity hotspots [9].

The close interaction between humans and animals in this region increases the risk of transmission of infectious agents from animals to humans causing zoonotic diseases. Worldwide, around 75% of new, so-called 'emerging' diseases in humans are zoonoses [10] while many others that are solely transmitted between humans today had their origins in animals [11]. Such diseases can have a devastating impact on public health and livelihoods. For example, Rift Valley fever (RVF)–a mosquito-borne zoonosis which causes recurring outbreaks in the

Horn of Africa—has contributed to thousands of human deaths due to haemorrhagic fever [7] and significant food insecurity owing to abortion storms and other production losses in livestock [12]. Further, subsequent bans on livestock export have had a significant impact on economies such as Somalia/Somaliland that are dependent on a single sector (livestock exports) and market (Saudi Arabia) [5,13]. While emerging zoonoses attract international attention and investment under the auspices of the Global Health Security Agenda (GHSA) [14], endemic zoonoses also pose a considerable threat to populations in low and middle income countries [15,16]. The Horn of Africa is particularly significant in this regard; in a global assessment supported by the United Kingdom Department for International Development (DFID), Kenya, Ethiopia, Sudan and Uganda were identified in the top 20 countries at the interface of poverty, emerging livestock systems and zoonotic disease burden [15].

Many endemic zoonoses fall into the category of neglected diseases which predominantly affect poor and marginalised populations and which neither attract the adequate health resources nor the research effort needed for their effective control [17]. Meanwhile they perpetuate poverty by attacking not only people's health but also their livelihood through reducing livestock productivity [17]. Of the twenty neglected tropical diseases prioritised by the Word Health Organisation (WHO), seven are recognised as neglected zoonotic diseases (NZD). This includes rabies and diseases caused by tapeworms such as echinococcosis and taeniasis/cysticercosis, which WHO recently proposed should be targeted for control in order to achieve the sustainable development goals (SDGs) [18]. Many countries in the Horn of Africa are highly afflicted by such diseases. For instance, Ethiopia ranks second among African countries with regards to total deaths caused by rabies, while Somalia ranks first in terms of rabies deaths per capita [19].

Investing in interdisciplinary approaches, including 'One Health', has been highlighted as one of ten recommendations to prevent as well as respond to zoonotic disease outbreaks and pandemics [20]. 'One Health' acknowledges that the health of humans and animals are interdependent and intricately linked to the health of the ecosystems in which they co-exist; it consequently encourages multidisciplinary and multisectoral collaborations to tackle complex issues at the human-animal-environment interface [21]. As such, 'One Health' approaches are particularly relevant to zoonoses research, and may become more important in the age of the Anthropocene [22] as related planetary impacts such as climate change are felt [23]. To improve the quality of 'One Health' research, Davis *et al.* [24] have developed the 'Checklist for One Health Epidemiological Reporting of Evidence (COHERE statement)', in which they suggest that studies reported to be 'One Health' in nature should present data on all three domains (human, animal, and environment).

Several countries in the Horn of Africa have conducted zoonoses prioritisation exercises based on a methodology developed by the United States Centers for Disease Control and Prevention [25–28]. Consistent with a 'One Health' approach, these multisectoral prioritisation workshops afford many benefits including subsequent collaborative work on prioritised zoonoses [29,30]. Nonetheless, they are limited by the lack of data on the burden of some zoonoses, especially NZD. Further, when data do exist, the quality of the study design and completeness of reporting often remains inadequate to appropriately inform policy and guide research effort, as was found in a recent review of zoonotic disease research on four livestock value chains in Africa [31]. In order to more efficiently focus future research efforts, this scoping review aimed to provide a detailed analysis of the last 100 years of zoonosis research in the Horn of Africa. Specifically, we aimed to address the following research questions:

1. What has been the disease and country focus of zoonoses research in the Horn of Africa?

2. To what extent has research addressed country priorities for zoonoses?

3. What data have been reported (human, animal or environment)?

4. What discipline/methods have been adopted?

5. Who has been doing the research (local/foreign researchers) and to what extent has collaboration been occurring within and between countries?

## Methods

We performed a scoping review of the literature available online following the PRISMA Extension for Scoping Reviews (PRISMA-ScR) guidelines (S1 Table) [32].

### Search strategy

The review was conducted using five online databases: PubMED, Web of Science, Scopus, Cab Direct and ProQuest Dissertations and Theses. Web of Science is an aggregate database that provides access to Arts & Humanities Citation Index, Science Citation Index Expanded, Social Sciences Citation Index and Conference Proceedings Citation Index—Science, as well as BIO-SIS Previews, Medline and Journal Citation Reports. Similarly, Cab Direct provides access to CAB abstracts and Global Health.

The search strategy encompassed two approaches (Fig 1 and S2 Table). First, we undertook disease-specific searches on the thirteen zoonoses ranked as 'most important' in the comprehensive DFID report based on impact on human health (mortality and morbidity), impact on livestock, amenability to interventions, severity of disease and emergence [15]. This includes: gastrointestinal infections (zoonotic), leptospirosis, cysticercosis, tuberculosis (zoonotic), rabies, leishmaniasis, brucellosis, echinococcosis, toxoplasmosis, Q fever, trypanosomiasis (zoonotic), anthrax and hepatitis E. Given the significance of Rift Valley fever to the Horn of Africa [33], it was also included in the disease-specific search. Consistent with the DFID report, we focussed on *Salmonella*, *Campylobacter*, toxigenic *Escherichia coli* and *Listeria* as the zoonotic gastrointestinal infections of interest. Disease-specific search strings were constructed using a combination of scientific disease name (e.g. cysticercosis), alternative name (e.g. 'pork tapeworm') and agent name (e.g. '*Taenia solium*'), combined using the Boolean operator, 'OR'. For diseases that have zoonotic and non-zoonotic transmission (i.e. leishmaniasis, trypanosomiasis, tuberculosis), disease names were combined with the term 'zoonosis OR zoonoses OR zoonotic' using the Boolean operator, 'AND', unless specific species could be used to denote zoonotic transmission (e.g. *Mycobacterium bovis*, *Trypanosoma brucei rhodesiense*). Second, we undertook a general search for literature containing 'zoonosis OR zoonoses OR zoonotic' in the title/abstract/keywords of papers which did not include the disease-specific search terms mentioned above. This enabled us to capture general literature on zoonoses, as well as those not deemed important in the DFID assessment. For this review we defined Horn of Africa as encompassing the following countries: Eritrea, Ethiopia, Djibouti, Kenya, Somalia, South Sudan, Sudan and Uganda. This corresponds to the eight-country trade bloc in Africa known as the Intergovernmental Authority on Development (IGAD). In recognition of the semi-autonomous nation of Somaliland, this term was also included as a search term, as was the general term, 'Horn of Africa'. Disease terms were combined with country terms using the Boolean operator, 'AND'. The final search terms are available in S2 Table. There was no language restriction at the database search level.

Database searches were performed within a day, on September 21st, 2018. Retrieved papers were subsequently included if they were in English or French, and published between 1918 and 2018 (i.e. 100 years). Records were exported to Endnote, v8.0.1 (Clavariate Analytics,

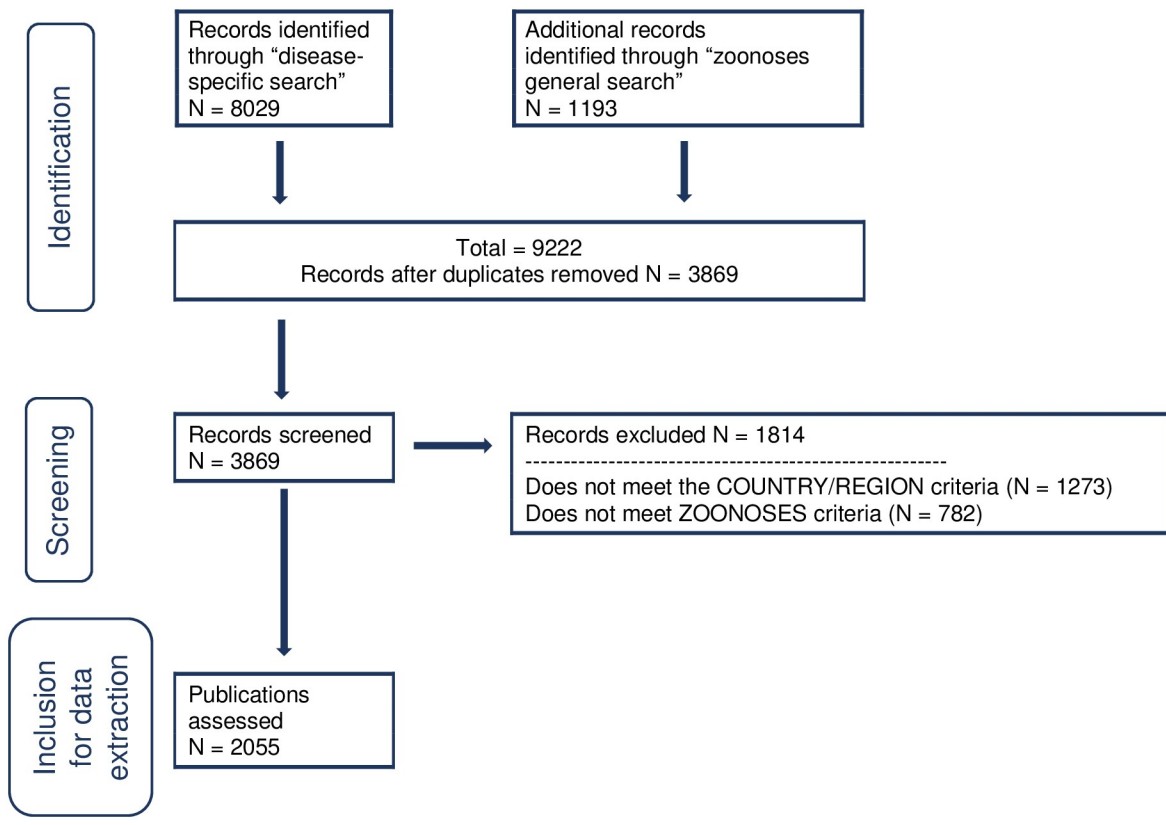

**Fig 1. Preferred Reporting Items for Systematic Reviews and Meta-Analyses (PRISMA) flow diagram detailing number of papers on zoonoses in the Horn of Africa retrieved and selected for data extraction.** Note that papers were excluded if they did not meet any one of the two inclusion criteria, hence numbers stated for each criterion do not add up to the total number of records excluded.

Philadelphia) [34], combined into one library and scanned for duplicates using methods described by Bramer *et al* [35]. Further de-duplication was performed in the online systematic review software, DistillerSR (Evidence Partners, Ontario, Canada) [36].

## Screening and data extraction

Screening and data extraction were performed in DistillerSR using a two-stage process. In the first stage, two reviewers independently screened the title and abstract of each paper and confirmed whether it met the eligibility criteria for geographic and zoonotic focus. Studies were included if they involved field data collection on zoonoses (outcome) in humans, domestic and/or wild animals, vectors or other environmental samples in a country of interest (population). Reviews and mapping studies were included if a country of interest was the main focus of the review (e.g. leptospirosis in Kenya), but excluded otherwise (e.g. leptospirosis in Africa). Laboratory studies involving isolates from a number of countries were included if the number of isolates from a country of interest exceeded half the total number of isolates and were excluded otherwise. Studies referring to migrants/travellers/imported animals from a country of interest were included if the country of interest was a clear focus (e.g. prevalence survey of cattle in Egypt following import from Sudan). For diseases with both zoonotic and non-zoonotic transmission, studies were only included if zoonotic transmission was the clear focus (e.g. refers to 'zoonoses', animals, or a zoonotic species such as *M. bovis*). This rule was

necessary to avoid including a large volume of papers referring to (for example) clinical research on tuberculosis in humans.

In the second stage, titles and abstracts were independently screened by two authors and data were extracted on the disease, country, type of data reported (human, animal, environment; 'One Health domains') and methods used for data collection and/or analysis for each paper. While Davis *et al.* [24] provide some definitions and examples of 'One Health domains', we found it necessary to refine the criteria in order to capture the full spectrum of data reported (S3 Table). In particular, we divided animal domain into studies involving assessment of whole animals/carcasses and/or those reporting evaluation of animal products. Where exposure to animals/animal products/environment were considered as risk factors for humans but no data was presented on such animals/animal products (e.g. questionnaire asking about exposure to cats in people with toxoplasmosis), studies were classified as human domain only unless criteria for other domains were also met (e.g. study documenting toxoplasmosis status in both humans and cats). Methods were classified in two steps. First, in large discipline categories, specifically: laboratory based/basic research, epidemiology, social sciences, environmental health/environmental sciences, economics, policy studies (for example zoonoses prioritisation workshops), and reviews. Second, more detailed methods for each category were specified (e.g. descriptive versus observational studies in epidemiology, focus group discussions (FGD) or quantitative surveys in social sciences). See full list of methods in S6 Table. Six investigators were involved in paper screening and data extraction. For both stages, to ensure consistency, conflicts in classification were resolved by one of the two main researchers (LC, SM) or by discussion in the case of disagreement. If needed, the full paper was retrieved and screened to resolve the conflict.

Finally, given the large number of papers, we used the easyPubMed package, v2.3 [37] in R, to automatically extract author affiliation data. Whilst this necessitated that the authorship analysis be limited to the smaller subset of papers obtained from PubMed, we do not anticipate that this would have introduced any specific bias.

## Data analysis

Data extracted in DistillerSR on country, disease, domain and method were exported in a.csv format and analysed using Microsoft Excel 2016 and R version 3.5.1 [38] in order to obtain summary tables and figures representing the number and proportions of publications grouped by country, disease, domain and discipline. We aggregated the reports for Sudan and South Sudan due to the inability to identify which Sudanese studies happened in South Sudan before it gained independence in 2011. Similarly, studies from Somalia and Somaliland were aggregated. Number of publications by disease and by country ("research effort") was compared to the rankings of zoonoses prioritised in the countries where such information was available, namely in Ethiopia (rabies, anthrax, brucellosis, leptospirosis and echinococcosis) [26]; Kenya (anthrax, trypanosomiasis, rabies, brucellosis and Rift Valley fever) [27]; and Uganda (anthrax, zoonotic influenza viruses, viral haemorrhagic fevers, brucellosis, African trypanosomiasis, plague, and rabies) [28]. Data were visualised using timeline graphs, radar charts and Venn diagrams; these were generated in R using ggplot, fmsb and eulerr packages, respectively [39–41].

Primary author affiliations (country and institute) were extracted from the affiliation data in Microsoft Excel 16. A unique identifier was created using a composite of the researcher's name and country of affiliation ("author-affiliation"). Thus, an author who had published with different country affiliations could have more than one unique author-affiliation. These data were transformed into networks in which nodes represented countries of authors' affiliation,

and links (edges) expressed collaborations (co-authorship) between countries. Collaboration was designated as occurring between authors in the same country ("intra-country") or between countries (international or "inter-country"). The size of each node was proportional to the number of authors who are affiliated with the country represented by the node. Links were weighted by the number of unique publications with shared authorship between two countries ("paper-collaborations"). Networks were generated for the whole of the Horn of Africa and related collaborations, as well as separate networks representing research focused on each of the countries of interest. Colours were used to differentiate collaboration type (e.g. Global North-Global South) and country economic classification (e.g. high income). Construction of networks and subsequent analyses were undertaken using R and the igraph package [42]. Maps and geographical networks were generated in Gephi [43] using GeoLayout and Map Of Countries layouts both under Apache v2 License.

## Results

A total of 9222 papers were retrieved after database searching (Fig 1). After removal of duplicates, the title and abstracts of 3869 papers were screened to verify compliance with the geographic and zoonoses criteria. Subsequently 1814 papers were excluded: 1273 did not meet the geographic criteria (1118 papers were outside of the geographic focus and information was missing for 155 papers) and/or 782 did not meet the zoonoses criteria (699 papers did not focus on a zoonotic disease and the information was missing for 83 papers). Thus, 2055 papers, published between 1938 and 2018, were deemed eligible and included for data extraction (See S4 Table for full list).

Overall, research effort on zoonoses in the Horn of Africa has increased over time, with a dramatic increase of publications in the 2000's; over 75% of the publications were published from 2000 onwards. This pattern is true for all countries except for Somalia, Eritrea and Djibouti which had sporadic publications on zoonoses (Fig 2 and see S1 Fig for details by country and by disease). We also note that research effort (measured by publication number) in Ethiopia seems to have reduced after a peak in 2013.

### Country and disease focus

The total number of publications by country and disease of focus is shown in Table 1. Overall, Ethiopia had the highest number of publications, followed by Kenya, Uganda and Sudan/South Sudan. Somalia/Somaliland, Djibouti and Eritrea had the smallest number of publications. Eleven publications had a regional (Horn of Africa) focus, eight of which focussed on RVF. Brucellosis, echinococcosis and gastrointestinal bacteria were the most studied zoonoses (40% of all publications), while anthrax, leptospirosis and Q fever were comparatively understudied. In addition to the fourteen specific zoonoses our search methodology focused on, over 50 'other zoonoses' were identified as a focus for 23% of the publications. The most common 'other zoonoses' were liver fluke (78 publications), cryptosporidiosis (44 publications), rickettsiosis (29 publications) and giardiasis (28 publications) (S5 Table).

Research effort on specific diseases differed by country although there were some consistent patterns (Table 1 and Fig 3). Research on brucellosis/echinococcosis, RVF/gastrointestinal bacteria and zoonotic trypanosomiasis dominated in Ethiopia, Kenya and Uganda, respectively. Brucellosis ranked in the top four for all countries in terms of number of publications. In contrast, some diseases such as anthrax, leishmaniasis, leptospirosis and Q fever were relatively under-studied across all countries. Overall, publications focusing on the published national priorities in Ethiopia, Kenya and Uganda represented only 45%, 32% and 52% of the publications of each country. Anthrax was a notable outlier; despite being prioritised in all 3

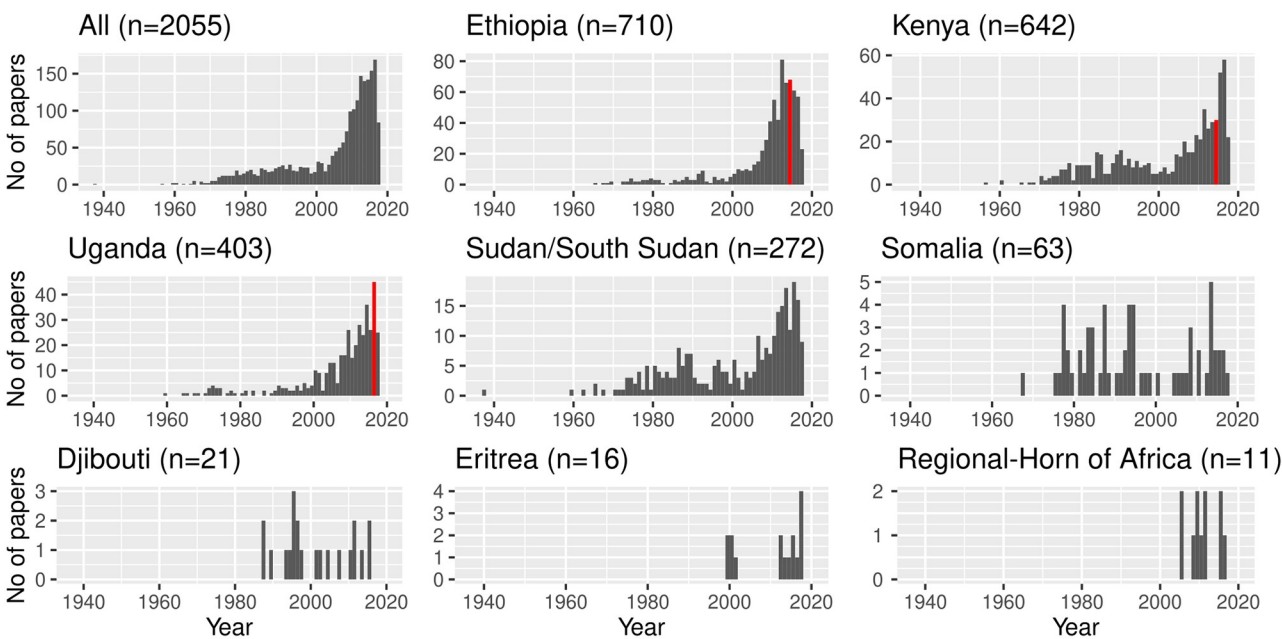

**Fig 2. Number of publications on zoonoses in the Horn of Africa, by country of focus and year.** The bars in red represent the year the national zoonoses prioritisation workshops were conducted for the concerned countries.

**Table 1. Number (No) of publications on zoonoses in the Horn of Africa, by country of focus and disease (n = 2055).** Note: the total number of publications and percentage do not total 100% as a single paper could report data on multiple diseases.

| Zoonoses | Ethiopia | Kenya | Uganda | Sudan / South Sudan | Somalia | Djibouti | Eritrea | Regional (Horn of Africa) | TOTAL* |
|---|---|---|---|---|---|---|---|---|---|
| | No (%) | No (%) | No (%) | No (%) | No (%) | No (%) | No (%) | No (%) | No (%) |
| Brucellosis | 101 (14) | 54 (8) | 61 (15) | 67 (25) | 17 (27) | 5 (24) | 7 (44) | 1 (9) | **306 (15)** |
| Echinococcosis | 127 (18) | 93 (14) | 14 (3) | 36 (13) | 6 (10) | 4 (19) | 1 (6) | 0 (0) | **272 (13)** |
| Gastrointestinal bacteria* | 78 (11) | 116 (18) | 38 (9) | 27 (10) | 7 (11) | 2 (10) | 1 (6) | 0 (0) | **256 (12)** |
| Tuberculosis** | 129 (18) | 14 (2) | 26 (6) | 8 (3) | 3 (5) | 5 (24) | 6 (38) | 1 (9) | **186 (9)** |
| Rift Valley fever | 4 (1) | 122 (19) | 16 (4) | 30 (11) | 10 (16) | 7 (33) | 1 (6) | 8 (73) | **183 (9)** |
| Trypanosomiasis** | 12 (2) | 45 (7) | 113 (28) | 10 (4) | 2 (3) | 0 (0) | 0 (0) | 0 (0) | **167 (8)** |
| Toxoplasmosis | 65 (9) | 19 (3) | 17 (4) | 22 (8) | 6 (10) | 5 (24) | 1 (6) | 0 (0) | **131 (6)** |
| Rabies | 77 (11) | 34 (5) | 8 (2) | 9 (3) | 3 (5) | 2 (10) | 1 (6) | 0 (0) | **126 (6)** |
| Cysticercosis/Taeniasis | 61 (9) | 39 (6) | 11 (3) | 6 (2) | 4 (6) | 1 (5) | 0 (0) | 0 (0) | **119 (6)** |
| Hepatitis E | 11 (2) | 2 (0) | 14 (3) | 19 (7) | 9 (14) | 2 (10) | 1 (6) | 0 (0) | **55 (3)** |
| Leishmaniasis | 19 (3) | 13 (2) | 2 (0) | 17 (6) | 2 (3) | 1 (5) | 0 (0) | 0 (0) | **53 (3)** |
| Anthrax | 26 (4) | 8 (1) | 5 (1) | 5 (2) | 2 (3) | 1 (5) | 0 (0) | 0 (0) | **47 (2)** |
| Leptospirosis | 5 (1) | 20 (3) | 7 (2) | 12 (4) | 4 (6) | 1 (5) | 0 (0) | 0 (0) | **47 (2)** |
| Q Fever | 8 (1) | 19 (3) | 2 (0) | 9 (3) | 3 (5) | 3 (14) | 0 (0) | 0 (0) | **41 (2)** |
| Other zoonoses | 123 (17) | 168 (26) | 119 (30) | 47 (17) | 20 (32) | 10 (48) | 2 (13) | 1 (9) | **463 (23)** |
| **TOTAL** | **710** | **642** | **403** | **272** | **63** | **21** | **16** | **11** | **2055** |

\* *Salmonella*, *Campylobacter*, toxigenic *Escherichia coli*, *Listeria*

\*\* Only when zoonotic transmission was considered

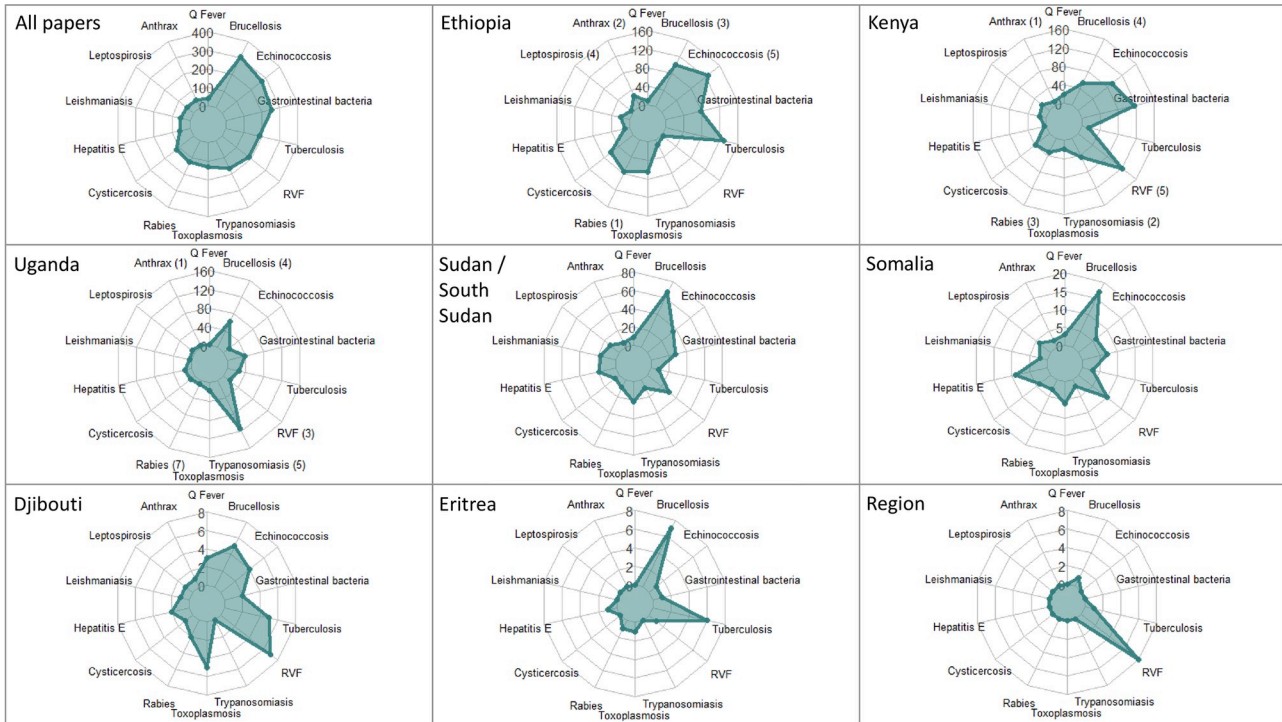

**Fig 3. Radar charts showing number of publications on zoonoses in the Horn of Africa, by country of focus and disease (n = 2055).** Numbers in parentheses (n) identify the rank of each zoonosis in national prioritisation workshops (when available). RVF = Rift Valley fever. Note: only those fourteen diseases included in the specific disease search are shown; for full listing of "Other zoonoses" see S5 Table.

countries, this disease was the focus of only 4%, 1% and 1% of publications in each country, respectively. Meanwhile, bovine tuberculosis in Ethiopia and gastrointestinal bacteria in Kenya had a high number of publications despite not being prioritised.

## Domain focus

Overall, two thirds of publications reported data on the animal domain while half of publications reported data on humans; only 14% reported data on the environment (Table 2). Studies on animals mainly involved livestock species (70%). Wild animals (18%; primarily non-human primates, but also bats and rodents) and cats and dogs (12%) were also studied. Amongst the publications reporting on the environment (n = 285), a majority (n = 211, 74%) considered "biotic" factors; 68% (n = 143) of such papers reported on free-living arthropod vectors, including 60 papers on RVF vectors (*Aedes* spp., *Culex* spp., etc.) and 34 on zoonotic trypanosomiasis vectors (*Glossina* spp.).

Three quarters of the publications reported on one domain only (animal, human or environment), while about 20% reported data in both animal and human domains. Only 4% (n = 73) of publications reported on all three 'One Health domains' (Fig 4). However, there was significant variation in the domains reported by disease. For example, papers on hepatitis E were skewed towards humans (95% included the human domain). In contrast, >70% of publications on anthrax, brucellosis, cysticercosis/taeniasis, echinococcosis, rabies and tuberculosis focused on animal cases. Leptospirosis and toxoplasmosis represent diseases with very little reporting of both human and animal domains, and very limited investigation of the environment domain despite the recognised role of environmental factors in the epidemiology of

**Table 2. Number of publications on zoonoses in the Horn of Africa, by data reported.** At the highest level, data were classified into the three 'One Health domains', namely human, animal and environment. Animal and environment domains were further sub-classified as shown. Note: the total number of publications and percentages do not add up to 100% as a single paper could report data on multiple domains.

| One Health domain | No. of publications (%) |
|---|---|
| **Human** | **1031 (50)** |
| **Animal** | **1321 (64)** |
| Livestock | 926 (70) |
| Wildlife & fish | 239 (18) |
| Cat/dog | 155 (12) |
| Animal products (meat, milk, egg) | 100 (8) |
| Laboratory rodent, rabbit, guinea pig, hamster | 45 (3) |
| On-host ectoparasites (fleas, ticks, lice, etc.) | 43 (3) |
| Not specified | 66 (5) |
| **Environment** | **285 (14)** |
| Biotic (free-living arthropods, plants, etc.) | 211 (74) |
| Abiotic (water, soil, climate, etc.) | 133 (47) |
| **Total** | **2055** |

these diseases. Vector-borne zoonoses like leishmaniasis, trypanosomiasis and RVF had the highest proportion of papers covering all three 'One Health domains' (15%, 13% and 8% respectively) and also an almost even reporting across them, with more than 30% of the publications reporting on each domain.

## Disciplines focus and methods

Table 3 shows the discipline focus and methods used within papers (see S6 Table for more detailed analysis). The majority (75%) of studies employed epidemiological methods, half of which were prevalence surveys or case reports/case series of zoonoses while the other half was mainly represented by cross-sectional studies evaluating risk factors for zoonoses (S6 Table). Only 2% of papers involved experimental studies such as randomised controlled trials. Basic scientific laboratory methods were used in about one third of the studies and mainly involved application of advanced taxonomic methods like genotyping. Social sciences methods were applied in only 11% of the papers, and mostly involved quantitative or semi-quantitative surveys like 'knowledge, attitude and practices' surveys. Environmental science/environmental health approaches were also applied in 11% of the papers, with a majority of these using entomology methods. Other methods (economics, opinion, review, policy) were used in a minority of publications.

While epidemiological methods dominated, there were notable variations by disease. For example, laboratory methods were frequently used for studies on gastrointestinal bacteria while laboratory aspects of anthrax, brucellosis and hepatitis E were relatively under-studied. Social science approaches were mostly applied in anthrax studies but were less utilised in hepatitis E, leishmaniasis, leptospirosis and Q fever research. Environmental sciences/environmental health approaches were mainly applied to RVF and Q fever (entomology) as well as gastrointestinal bacteria (environment and food testing). Economic methods were often used to estimate the financial losses caused by cysticercosis/taeniasis and echinococcosis due to carcass condemnation at slaughterhouses.

Overall two thirds of the publications addressed a single discipline. The diseases with the highest share of multidisciplinary papers focused on gastrointestinal bacteria (63%),

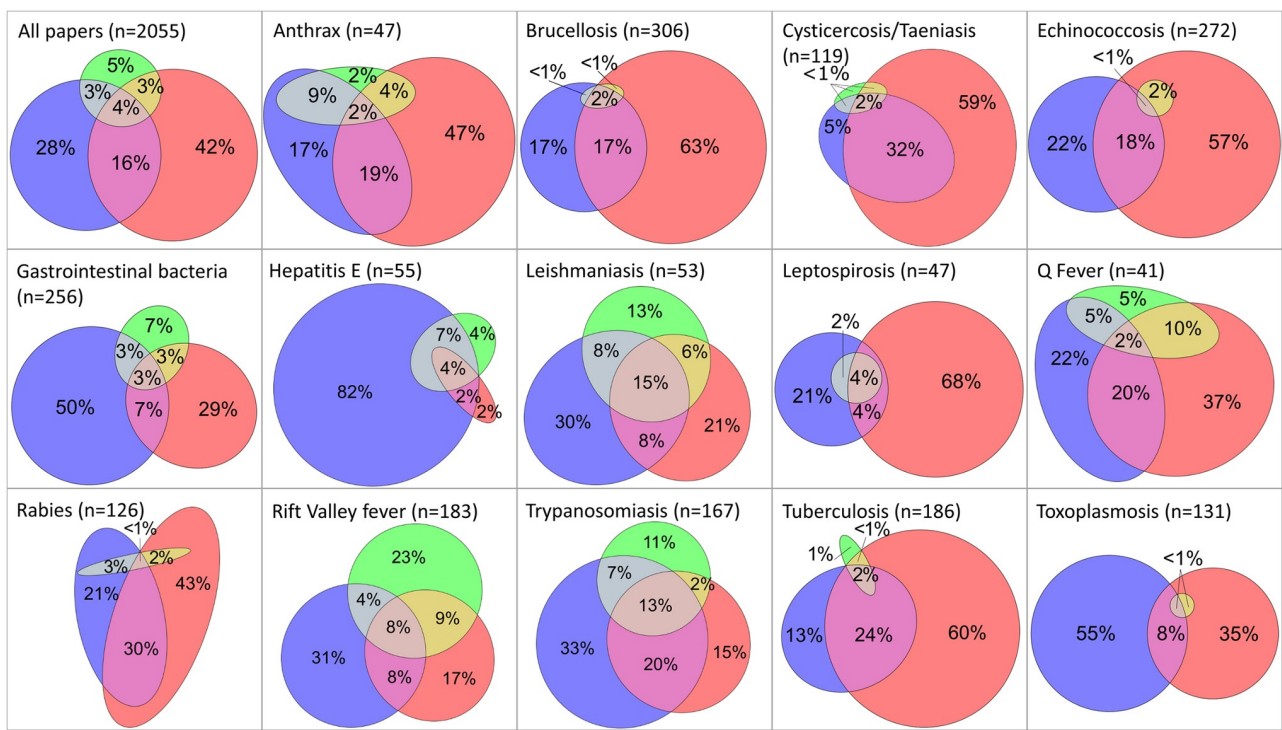

**Fig 4. Venn diagrams showing the data reported in 2055 publications on zoonoses in the Horn of Africa, by disease and 'One Health domain'.**
Note: blue is for human, red for animal and green for environment domain.

cysticercosis/taeniasis (55%) and echinococcosis (47%), primarily because such studies tended to employ field-based epidemiological sampling and laboratory and/or economics methods (Table 3 and S6 Table).

## Authorship analysis

A total of 1071 papers, representing 5059 unique author-affiliations and 2230 paper-collaborations were included in the sub-analysis of authorship patterns (see Table 4). Overall, just under half of the unique author-affiliations (48%) were in the country of focus. Ethiopia was the only country of focus where a majority of author-affiliations (57%) were inside this country (local). In contrast, more than two thirds of unique author-affiliations associated with research in Uganda and Sudan/South Sudan were outside of these countries (foreign); this fraction increased even further in Somalia/Somaliland, Djibouti and Eritrea. A majority of authors with foreign affiliations were associated with an institution in the United States or United Kingdom (18% and 10% of all unique author-affiliations, respectively) (Fig 5). Overall, 57% of 2230 paper-collaborations involved researchers within the same country (intra-country collaboration), a substantial proportion (41%) of which involved authors affiliated with a country in the Global North collaborating with other authors from the same country. This was particularly true for research focussing on Uganda and Somalia/Somaliland. Of 965 paper-collaborations that involved international (inter-country) collaboration, 57% involved Global North-Global South collaboration, while only 30% involved Global South-Global South collaboration. Global South-Global South collaboration was highest for research focussing on Sudan/South Sudan (44% of paper-collaborations), Kenya (39%) and Ethiopia (38%). Collaborations involving only researchers from the Global-South were scattered across 41 countries with Kenya-

**Table 3. Disciplinary/methodological approaches described in publications on zoonoses in the Horn of Africa, by disease (n = 2055).** Note: numbers represent the percentage of the total number of papers that used each discipline/method. Intensity of colour shading is reflective of the proportion of papers using each approach for a given disease. Percentages do not add up to 100% as a single paper could involve multiple disciplinary/methodological approaches.

| Diseases | Disciplinary/methodological approach (% of papers) | | | | | | | | | No of disciplines/ paper (% of papers) | | |
|---|---|---|---|---|---|---|---|---|---|---|---|---|
| | Epidemiology | Laboratory | Social sciences | ES—EH** | Review | Economics | Policy | Opinion | Other | 1 | 2 | >2 |
| All diseases | 75 | 31 | 11 | 11 | 5 | 4 | 1 | 1 | 0 | 66 | 29 | 5 |
| Anthrax | 64 | 6 | 34 | 6 | 9 | 0 | 6 | 0 | 2 | 72 | 28 | 0 |
| Brucellosis | 90 | 8 | 14 | 6 | 5 | 2 | 1 | 0 | 0 | 79 | 17 | 4 |
| Cysticercosis/Taeniasis | 86 | 38 | 16 | 2 | 8 | 21 | 0 | 1 | 1 | 45 | 43 | 13 |
| Echinococcosis | 83 | 42 | 9 | 1 | 5 | 18 | 0 | 1 | 0 | 53 | 34 | 13 |
| Gastrointestinal bacteria | 79 | 62 | 5 | 23 | 2 | 0 | 1 | 0 | 0 | 37 | 55 | 8 |
| Hepatitis E | 82 | 4 | 7 | 9 | 13 | 2 | 0 | 0 | 0 | 84 | 16 | 0 |
| Leptospirosis | 81 | 15 | 2 | 2 | 4 | 0 | 2 | 0 | 2 | 77 | 21 | 2 |
| Leishmaniasis | 62 | 43 | 4 | 11 | 17 | 0 | 0 | 4 | 0 | 79 | 21 | 0 |
| Q fever | 73 | 12 | 10 | 22 | 7 | 2 | 0 | 5 | 0 | 73 | 22 | 5 |
| Rabies | 53 | 19 | 29 | 6 | 5 | 3 | 4 | 6 | 2 | 79 | 17 | 4 |
| Rift Valley fever | 61 | 10 | 8 | 25 | 4 | 3 | 1 | 3 | 0 | 85 | 14 | 1 |
| Toxoplasmosis | 89 | 12 | 7 | 0 | 5 | 1 | 0 | 1 | 0 | 88 | 11 | 2 |
| Trypanosomiasis* | 53 | 38 | 11 | 10 | 7 | 2 | 1 | 2 | 1 | 77 | 20 | 3 |
| Tuberculosis * | 76 | 43 | 18 | 3 | 4 | 2 | 1 | 3 | 1 | 55 | 41 | 4 |

* Only when zoonotic transmission was considered

** Environmental science and environmental health

South Africa (4%), Kenya-Uganda (3%) and Kenya-Ethiopia (3%) representing the most common paper-collaboration pattern. A full listing of the nodes and links included in the networks is available in S1 File.

## Discussion

We present, to the best of our knowledge, the most comprehensive scoping review of research on zoonoses in the Horn of Africa, identifying strengths, weaknesses and gaps in the existing body of literature. We assessed 2055 eligible publications on over 60 different zoonoses, in seven countries of the Horn of Africa. Our results corroborate and expand upon the findings of a recent review on zoonoses research in East Africa [44] which focussed only on Ethiopia, Kenya and Uganda. We found a growing trend in publications describing endemic zoonoses in the region. We also found that research efforts are often duplicated and do not always align well with country priority zoonotic diseases as identified in multi-stakeholder workshops. Despite zoonoses being a clear target for One Health research, a very limited proportion of studies report data on the three domains of human, animal and environment. We observed the dominance of descriptive and observational epidemiological studies, with a low proportion of multidisciplinary publications. Finally, we found that international collaborations were mostly Global North-Global South with a high proportion of authors affiliated with countries outside the Horn of Africa, particularly the Global North.

Nearly half of the papers in our review focused on just four diseases, namely: brucellosis, echinococcosis, gastro-intestinal bacterial zoonoses and bovine tuberculosis, although the disease focus differed by country. Brucellosis, a World Organisation for Animal Health (OIE)-

**Table 4. Institutional affiliations and type of collaboration of authors of publications on zoonoses in the Horn of Africa, by country of focus (n = 1071).** Number of unique author-affiliations and number of papers by nature of collaboration corresponds to node size and link thickness in Fig 5, respectively. Percentages are rounded up to the nearest whole number.

| | Country of focus | | | | | | | |
| --- | --- | --- | --- | --- | --- | --- | --- | --- |
| | Ethiopia | Kenya | Uganda | Sudan/ South Sudan | Somalia/ Somaliland | Djibouti | Eritrea | All*** |
| | No (%) | No (%) | No (%) | No (%) | No (%) | No (%) | No (%) | No (%) |
| **No. papers** | **331** | **350** | **263** | **117** | **24** | **9** | **13** | **1071** |
| **No. unique author-affiliations*** | **1447** | **1899** | **1322** | **663** | **178** | **51** | **48** | **5059** |
| Local affiliation (primary affiliation in the country of focus) | 822 (57) | 909 (48) | 412 (31) | 214 (32) | 13 (7) | 12 (24) | 12 (25) | 2429 (48) |
| Foreign affiliation (primary affiliation outside of the country of focus) | 625 (43) | 990 (52) | 910 (69) | 449 (68) | 165 (93) | 39 (76) | 36 (75) | 2630 (52) |
| **No. paper-collaborations**** | **884** | **1010** | **538** | **631** | **52** | **14** | **37** | **2230** |
| Intra-country collaboration (≥2 authors from the same country) | 399 (45) | 448 (44) | 317 (59) | 165 (26) | 30 (58) | 10 (71) | 15 (41) | 1265 (57) |
| All authors affiliated with an institution in the country of focus | 241 (60) | 225 (50) | 121 (38) | 60 (36) | 2 (7) | 3 (30) | 6 (40) | NA |
| All authors affiliated with an institution in another country in the Global South | 38 (10) | 44 (10) | 25 (8) | 46 (28) | 9 (30) | 3 (30) | 1 (7) | 746 (59) |
| All authors affiliated with an institution in a country in the Global North | 120 (30) | 179 (40) | 171 (54) | 59 (36) | 19 (63) | 4 (40) | 8 (53) | 519 (41) |
| Inter-country collaboration (≥2 authors from different countries) | 485 (55) | 562 (56) | 221 (41) | 466 (74) | 22 (42) | 4 (29) | 22 (59) | 965 (43) |
| Global South-Global South | 186 (38) | 219 (39) | 61 (28) | 204 (44) | 2 (9) | 1 (25) | 3 (14) | 285 (30) |
| Global North-Global South | 243 (50) | 290 (52) | 127 (57) | 213 (46) | 9 (41) | 3 (75) | 12 (55) | 550 (57) |
| Global North-Global North | 56 (12) | 53 (9) | 33 (15) | 49 (11) | 11 (50) | 0 (0) | 7 (32) | 130 (13) |

* An author might have more than one country of primary institutional affiliation, for example, if they moved institute.

** Multiple author papers only. A paper that has two authors with primary affiliations in different countries is equivalent to one paper-collaboration (i.e. link is present between two countries in Fig 5). Thus a paper involving ≥3 authors with primary affiliation in ≥3 different countries will count for >1 paper-collaboration.

*** In this case, affiliation in the country of study means affiliation to any of the countries in the Horn of Africa

listed disease with important impacts on live animal trade, has been identified as a priority disease in multi-sectoral workshops in Ethiopia [26], Kenya [27] and Uganda [28] and was the focus of numerous studies in all countries. This is likely facilitated by the availability of simple and affordable diagnostic tests making its detection easier [45,46]. However, most studies focussed on animals, with human-related aspects comparatively understudied. Anthrax is also regarded as a priority zoonotic disease in the three countries, however it has received very limited research attention across the region. As an outbreak prone disease that requires more specialised laboratory capacity to confirm, the study of anthrax may be more challenging than other priority zoonoses. Nonetheless anthrax is considered by farmers as one of the most important livestock diseases in the Horn of Africa [47–49] and likely warrants more research investment on livelihood impacts and feasible control strategies. Similarly, diseases like Q fever and leptospirosis have received less research attention. These diseases are commonly associated with febrile illness in people, but are challenging to diagnose and as such they are often misdiagnosed and treated as malaria in sub-Saharan Africa [50,51]. This potentially contributes to underestimation of their public health and economic significance [52,53] and may explain their lack of prioritisation in multi-sectoral ranking exercises. It would seem prudent

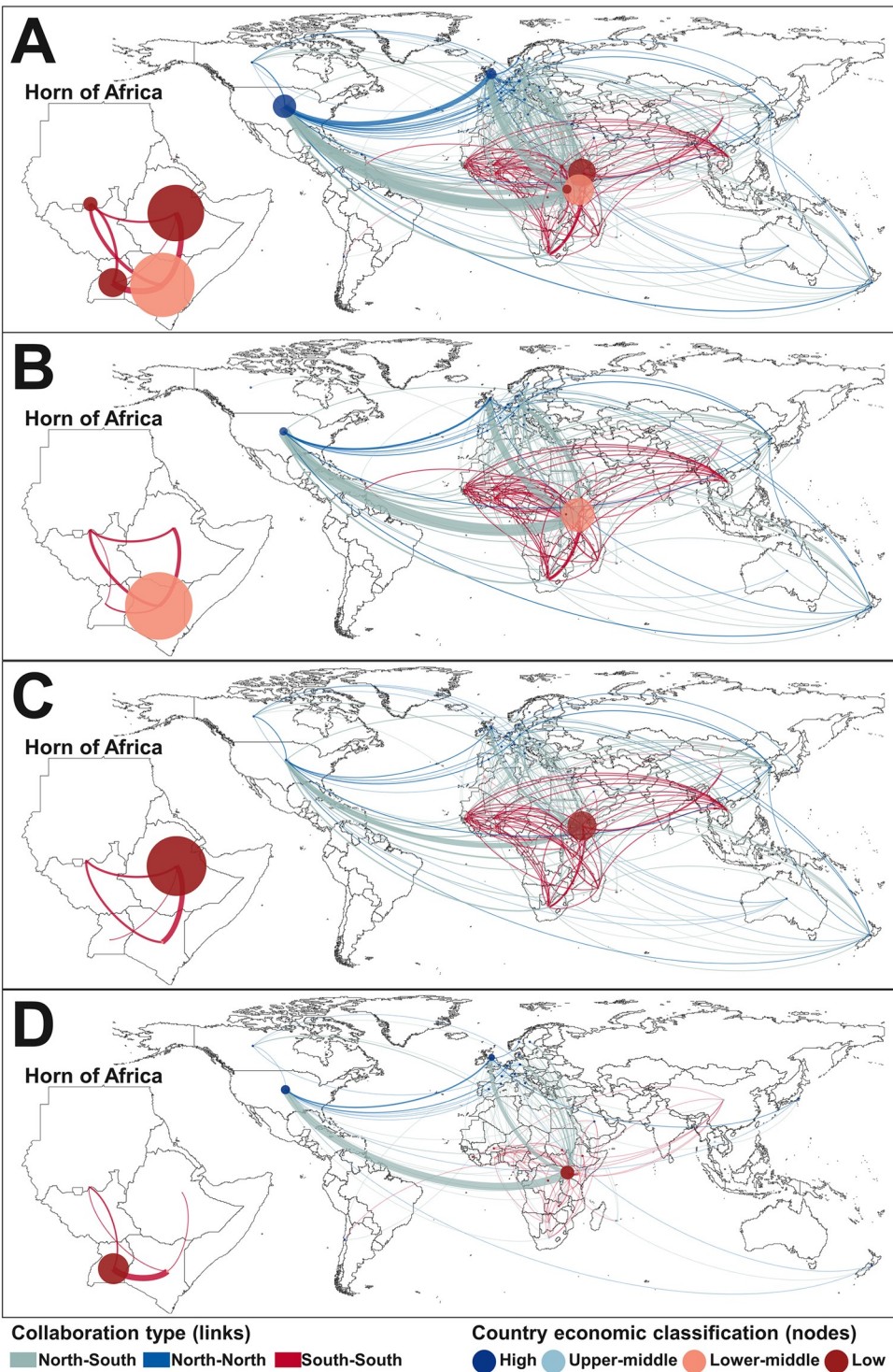

**Fig 5. International authorship network associated with publications on zoonoses in the Horn of Africa, by country of focus: (A) all papers, (B) Kenya, (C) Ethiopia, (D) Uganda.** In each panel, the regional collaboration within the Horn of Africa is shown as an inset. Node size and link thickness correspond to the number of unique author-affiliations and paper-collaborations in Table 4, respectively. As we aggregated papers from Sudan/South Sudan, this node is shown at the frontier between the two countries. Intra-country collaborations are not shown. The scale is the same across all maps. The base layer of the map is accessible here: https://datacatalog.worldbank.org/dataset/world-bank-official-boundaries. See S1 File for complete list of nodes and links shown in this figure.

to develop harmonised tools to comprehensively evaluate the societal burden of zoonoses in order to inform national prioritisation exercises independent of the existing publication bias [54].

We examined the domains reported in publications as a way to gauge the degree to which studies on zoonoses might be characterised as 'One Health' according to definitions established in the COHERE checklist [24]. While there was some variation by disease, we found that less than one in twenty research papers on zoonoses in the Horn of Africa simultaneously reported data on human, animal and environmental domains. Examples of such papers included: RVF and hepatitis E outbreak investigation [55,56], antimicrobial resistance profiles of *Salmonella* from different sources [57,58], risk factor studies of cysticercosis and Q fever [59,60], and policy papers on zoonoses prioritisation or decision support [61,33]. It is possible that this underestimates the true extent of joint data collection given researchers often encounter difficulties publishing multi/transdisciplinary work as a single paper [24]. We also feel that the COHERE checklist might be too restrictive in its definition of 'One Health' research as it applies this label strictly to papers reporting directly or indirectly on all three domains. This is especially limiting when applied to studies on zoonoses with direct animal-human transmission where the environment plays a limited role (e.g. rabies). Given rabies is widely acknowledged as a 'One Health' priority in many countries [26–28], the COHERE checklist seems inconsistent with actual practice. In any case, with the publication of the COHERE checklist, increasing recognition of the 'One Health' concept and related investments in the Africa region and elsewhere [62–64], we expect that the fraction of papers simultaneously reporting data from humans, animals and the environment will grow.

The majority of research on zoonoses in the Horn of Africa has used descriptive epidemiological methods (lower on the evidence pyramid; [65]) with comparatively little research using multidisciplinary or other methods, such as from the social sciences and humanities. This was also highlighted in the previous review of zoonoses research in East Africa [44]. Where multidisciplinary methods were used, we found this was often in the context of slaughterhouse studies on echinococcosis and taeniasis/cysticercosis wherein a combination of epidemiology, laboratory science, and economics methods were applied to measure the financial cost of carcass condemnation (e.g. [66–70]). Where social science studies did exist, these were mainly represented by quantitative surveys, often called "knowledge, attitude and practices" or KAP studies (e.g. [71–76]). KAP surveys emerged in the 1950s as a cost effective alternative to more in-depth social science methods and were designed to evaluate determinants and changes in behaviour in relation to a specific public health intervention [77–79]. In our experience in the zoonoses field, these studies are often undertaken by veterinary epidemiologists and rarely involve evaluation of an intervention. Taken as a whole our findings suggest greater investment is needed in research methods that generate higher-level evidence (e.g. randomised trials) with involvement of social sciences and humanities researchers to better understand the cultural context in which zoonoses are occurring [80]. Transparent zoonosis research agendas, such as the one by Steele *et al.* [81], should also be formulated and funded for the region to avoid the high repetition of similar descriptive epidemiologic studies within and across countries and encourage the development and evaluation of cost-efficient control solutions for prioritised diseases. Given the proliferation of KAP studies in this area, developing a specific framework to apply to zoonoses would also help to ensure quality and comparability of these reports.

Analysis of authorship patterns revealed the dominance of foreign scientists, particularly from the Global North, in zoonoses research in the Horn of Africa. This reflects a broader trend in research; despite comprising 12.5% of the world's population, Africa accounted for less than 1% of global research outputs in 2018 [82]. While the United States and the United

Kingdom rank respectively first and second globally in terms of medical science research outputs, the countries in the Horn of Africa ranked between 57 and 211 [83]. This likely translates to a considerable amount of foreign-led research output in the region. In a recent bibliometric analysis on Ebola research for example, the United States generated the greatest level of research output despite this disease occurring almost exclusively in Africa [84]. Further, similar to our study which showed limited Global South-Global South collaboration, other bibliometric studies also found little co-authorship between African researchers, with preference being given to collaboration with researchers in higher income nations [85].

Although we did not extract data on funding sources, we suspect these contribute substantially to the observed authorship pattern, with historical links between countries driving ongoing development assistance. Africa's gross expenditure on research and development as a proportion of GDP stands at about 0.5% compared to the world average of 2.2% [86]. In the recent decade, there has been substantial investment in zoonoses research and programming stimulated by DFID and United States Agency for International Development (USAID) through the 'Zoonoses in Emerging Livestock Systems' (ZELS) and 'Emerging Pandemic Threats' (EPT) initiatives, respectively. While international initiatives and financial support from high income countries remain critical for the development of research in the Horn of Africa, there is a growing emphasis on building more sustainable and equitable international research partnerships. Authorship as well as sample ownership and sample export have been identified as some of the key domains for equity in international health collaboration [87] and whilst contributions of team-members from the Global South might have been overlooked in the past, international guidelines should now be followed to ensure contributions of all local collaborators are duly recognised [88]. Furthermore, new partnerships should focus on local priorities and foster better balanced research output by empowering African researchers to contribute to and lead the design and conduct of projects [87,89–92]. Given similar disease ecosystems, livestock trade routes and value chains, it would seem reasonable to foster more intraregional collaborations between African scholars for zoonoses research in particular [89].

While this study presents one of the most comprehensive reviews on zoonoses research in any African region, several limitations must be mentioned. Firstly, we did not include theses and grey literature, except those published in the ProQuest Dissertations and Theses database. This likely contributed to under-estimation of the true research output on zoonoses in the Horn of Africa, especially by locally-affiliated researchers. Nonetheless, based on the findings of the review of zoonoses research in East Africa (which included publications and theses from Kenya only [44]), we expect the omission of theses to not substantially alter the findings reported here in terms of relative output by diseases, country, method and domains. Secondly, our search strategy specifically focused on fourteen zoonoses deemed to be of most relevance to poor livestock keepers in the Horn of Africa region [15]. While publications on other zoonoses would have been retrieved through the generic "zoonoses" search strategy, we acknowledge that we will have missed some publications on diseases that were not searched by name. This is particularly true for emerging diseases such as Ebola and Middle East Respiratory Syndrome which are also important in the region. Thirdly, a number of countries/states in the Horn of Africa have undergone varying degrees of changes with regard to internationally-recognised borders and this may have contributed to some country misclassification when we reported aggregated results. Furthermore, during data extraction, classification of papers in terms of epidemiological study design proved difficult due to mixed methods and often poorly described methodology, an issue previously identified in the veterinary literature [93]. While the two lead researchers, both epidemiologists (LC, SM) reached agreement on the classification of methods in each paper, it is possible that other researchers may have reached somewhat different conclusions. While COHERE also suggests 'One Health' research should incorporate

expertise from each domain [24], this proved difficult to do based on institutional affiliation only. Engagement of stakeholders and members of the community is also considered a feature of 'One Health' approaches [94,95], although we did not assess this. Finally, we may have over-estimated the number of foreign author-affiliations; for example, a number of international organisations (e.g. US Centers for Disease Control and Prevention) have regional offices which employ local staff, even if the affiliation is stated as overseas.

## Conclusion

There is a growing interest in zoonoses research in the Horn of Africa with over 2055 publications available on this topic. To sustainably improve human population's health and livelihoods, future research efforts in the region should ensure that:

1.  A 'One Health' approach that is based on holistic, transdisciplinary methods and follows high quality research standards is adopted (COHERE, PRISMA, etc.);

2.  National (priority zoonoses) as well as global priorities (SDGs) are addressed; and

3.  Local researchers are engaged in the development and implementation of projects with support of regional and international partnerships.

Establishing national and regional research agendas on zoonoses and mapping national and international funding accordingly would help to ensure these goals are met.

## Supporting information

**S1 Table. Preferred Reporting Items for Systematic reviews and Meta-Analyses extension for Scoping Reviews (PRISMA-ScR) Checklist (adapted from Tricco *et al*. 2018 [32]).** (DOCX)

**S2 Table. Final search terms.** (DOCX)

**S3 Table. Criteria used to define which 'One Health domains' are covered by a publication.** (DOCX)

**S4 Table. List of papers (n = 2055) included in the study with their references.** (XLSX)

**S5 Table. Number of publications per disease and per country.** Note: the total number of publications and percentages do not add up to 100% as a single paper could report data on multiple diseases. CCHF = Crimean-Congo Haemorrhagic Fever. (XLSX)

**S6 Table. Proportion of publications based on disciplinary/methodological approaches employed.** Note: the total number of publications and percentages do not add up to 100% as a single paper could report on more than 1 discipline/method. FGD = Focus group discussion. (XLSX)

**S1 Fig. Number of publications on zoonoses by year and disease, across all countries of the Horn of Africa (A) and in Ethiopia (B), Kenya (C), Uganda (D), Sudan/South Sudan (E), Somalia (F), Djibouti (G), Eritrea (H) and those with a regional focus (I).** Note: The bars in red represent the year the concerned zoonoses were prioritized in respective national workshops. In cases where no publications were produced that year, the year is indicated by a star.

A solid light grey plot means there is no publication for a given disease in this country. (DOCX)

**S1 File. Table presenting the link weight and node size of all networks presented in Fig 5.** (XLSX)

## Author Contributions

**Conceptualization:** Lisa Cavalerie, K. Marie McIntyre, Robert Christley, Gina Pinchbeck, Matthew Baylis, Siobhan M. Mor.

**Data curation:** Lisa Cavalerie, Ophélie Lebrasseur, Mark Nanyingi, K. Marie McIntyre, Matthew Baylis, Siobhan M. Mor.

**Formal analysis:** Lisa Cavalerie, Maya Wardeh, Siobhan M. Mor.

**Funding acquisition:** Daniel Asrat, Robert Christley, Gina Pinchbeck, Matthew Baylis.

**Investigation:** Lisa Cavalerie, Maya Wardeh, Ophélie Lebrasseur, Mark Nanyingi, K. Marie McIntyre, Matthew Baylis, Siobhan M. Mor.

**Methodology:** Lisa Cavalerie, Mark Nanyingi, K. Marie McIntyre, Robert Christley, Gina Pinchbeck, Matthew Baylis, Siobhan M. Mor.

**Project administration:** Lisa Cavalerie, Siobhan M. Mor.

**Supervision:** Siobhan M. Mor.

**Validation:** Lisa Cavalerie, Siobhan M. Mor.

**Visualization:** Lisa Cavalerie, Maya Wardeh, Siobhan M. Mor.

**Writing – original draft:** Lisa Cavalerie, Maya Wardeh, Siobhan M. Mor.

**Writing – review & editing:** Lisa Cavalerie, Maya Wardeh, Ophélie Lebrasseur, Mark Nanyingi, K. Marie McIntyre, Mirgissa Kaba, Daniel Asrat, Robert Christley, Gina Pinchbeck, Matthew Baylis, Siobhan M. Mor.

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
