## [Decision Letter · Decision Letter 0]

27 Apr 2021

Dear Dr Cavalerie,

Thank you very much for submitting your manuscript "One hundred years of zoonoses research in the Horn of Africa: a scoping review" for consideration at PLOS Neglected Tropical Diseases. As with all papers reviewed by the journal, your manuscript was reviewed by members of the editorial board and by several independent reviewers. The reviewers appreciated the attention to an important topic. Based on the reviews, we are likely to accept this manuscript for publication, providing that you modify the manuscript according to the review recommendations. 

Dear authors: Thank you for your submission to PLoS NTD. My sincere apologies for the long review process. Given the demands of the current pandemic on reviewers time and energy we struggled to find suitable reviewers with time available. Despite those challenges, generally the reviewers found your manuscript well written and have a few minor additions and considerations that could increase the impact of your review. I encourage you to look carefully at the comments and address their queries and suggestions. We look forward to receiving your revised manuscript. Yours, Brian Bird UC Davis One Health Institute and PLoS Associate Editor.

Sincerely,

Brian Bird, DVM, ScM, PhD

Associate Editor

Stuart Blacksell

Deputy Editor

Dear authors: Thank you for your submission to PLoS NTD. My sincere apologies for the long review process. Given the demands of the current pandemic on reviewers time and energy we struggled to find suitable reviewers with time available. Despite those challenges, generally the reviewers found your manuscript well written and have a few minor additions and considerations that could increase the impact of your review. I encourage you to look carefully at the comments and address their queries and suggestions. We look forward to receiving your revised manuscript. Yours, Brian Bird UC Davis One Health Institute and PLoS Associate Editor.

Reviewer's Responses to Questions

**Key Review Criteria Required for Acceptance?**

**Methods**

-Are the objectives of the study clearly articulated with a clear testable hypothesis stated?

-Is the study design appropriate to address the stated objectives?

-Is the population clearly described and appropriate for the hypothesis being tested?

-Is the sample size sufficient to ensure adequate power to address the hypothesis being tested?

-Were correct statistical analysis used to support conclusions?

-Are there concerns about ethical or regulatory requirements being met?

Reviewer #1: Study objectives, study design, and analyses were carried out sufficiently in accordance with PRISMA guidelines. No concerns around ethical or regulatory requirements.

Reviewer #2: The methods are overall clearly stated and their limitations also openly described in the discussion. I have only minor comments:

-Please state if there were any language restrictions made in the search

-Please state the R version used

**Results**

-Does the analysis presented match the analysis plan?

-Are the results clearly and completely presented?

-Are the figures (Tables, Images) of sufficient quality for clarity?

Reviewer #1: Presented analyses match the outlined analysis plan. Results are clearly presented and support the conclusions being discussed. Figure 4 is difficult to see. The nodes could be more clearly defined, as could the link thickness.

Reviewer #2: - The results are rather thorough and matching each research objective.

- I was hoping however, while reading through the results, to see a figure with publications over time. The figures picturing publications overtime are presented in details in the appendix, but it would be nice to also have an overall one in the main manuscript (I would suggest publications over time per country, not necessarily per disease in the main text). On the figures showing number of publications over time, I would add an arrow for when the national list of priority zoonoses was established. That would allow the reader to reflect on your comment on how research on zoonotic diseases does not always fit the list of priority zoonoses established by the government, in light of when one happened compared to the other. Maybe we would notice an increase of publications on a priority disease after official prioritization, maybe not, but the info is missing.

- It is not clear to me why only 1071 papers out of 2055 were included in the authorship analysis. Can the authors clarify in the text?

**Conclusions**

-Are the conclusions supported by the data presented?

-Are the limitations of analysis clearly described?

-Do the authors discuss how these data can be helpful to advance our understanding of the topic under study?

-Is public health relevance addressed?

Reviewer #1: The conclusions are supported by data presented. The points I would wish for the authors to expand upon are outlined in the Summary and General Comments below.

Reviewer #2: The discussion and conclusions are relevant and supported by the data presented.

**Editorial and Data Presentation Modifications?**

Reviewer #1: See Summary and General Comments.

Reviewer #2: (No Response)

**Summary and General Comments**

Reviewer #1: In this piece, authors present a comprehensive scoping review of zoonoses research in the Horn of Africa. This is well-done and thorough review, which answers important questions existing about the nature of interdisciplinary collaboration and reporting,, which is highly relevant to the One Health reporting guidelines and One Health frameworks being pushed forward in recent years. I appreciate the authors’ interpretation of their data, and I would only request discussing a few additional points (outlined below).

In addition to empowering local researchers to engage in zoonoses research, I would encourage the discussion to touch on the responsibility of authors from Northern countries to ensure authorship is inclusive of all field members on a team. I suspect contributions from authors in the study area are underrepresented and increasing dialogue within the research community about the importance of acknowledging work from all in-country partners is long overdue; your manuscript provides an opportunity to touch on this important point.

While it is illustrated in Figure 4, it would be nice if you could verbally expand upon, or share, the list of countries represented in the South-South collaborations. These are indeed more rare than South-North collaborations, and I think these relationships are worth exploring, particularly as it relates to your other points about out-of-country collaboration shaping research landscapes and priorities. Along this same line, when comparing those diseases that are locally prioritized (but not considered global priority), did you see a significant difference in author origin (e.g. local/in-country only vs. involvement of foreign/Global North collaborators)? This relates to your commentary surrounding the involvement of Global North authors shaping the research landscape (line 470), and I fully agree. If it’s within the scope of the review, or perhaps as a follow-up commentary, suggestions and directions for improving representation of local in-country authors would be beneficial. For instance, how underrepresented are in-country others on authorship lists? I couldn’t agree with your statement in lines 486-488 more.

Please describe inclusion/exclusion criteria for biotic environmental factors more specifically. This is particularly important owing to the number of vector-borne diseases covered in this scoping review. Please break down the 68% of biotic papers reporting on free-living arthropod vectors to investigate which arthropods were most reported on. I’m a bit surprised there aren’t more papers listed here, considering the burden of vector-borne disease in this region. It’s worth teasing this apart a bit, if the editor feels it is within the scope of this paper.

Lines 428-432: Could you expand upon how VBD having higher degree of tri-sector reporting indicates the COHERE checklist is too restrictive? This is an interesting point and worth going into a bit more detail on, since not all readers may be familiar with the specifics of the COHERE checklist.

The authors discuss the importance of reporting on all three sectors/domains (humans, animals and the environment), which certainly would help frame the results of studies through a more holistic lens and help check off the COHERE guidelines. However, reporting on all three domains often requires additional resources or additional investigation and analyses that may be cost and time-prohibitive. I wonder if there would be a way to anecdotally describe patterns you appreciated in those papers that DID report on all three domains, citing and referencing these papers specifically as a means of encouraging other authors to pursue similar frameworks. This may be a more tangible and inspirational example than pointing authors aspiring to report through a One Health framework to the COHERE guidelines.

Reviewer #2: I do not have further comments. The manuscript is very well written, the authors are transparent about their methods and results, and these results are needed to highlight solutions for a way forward in One Health research especially in the global South.

PLOS authors have the option to publish the peer review history of their article (what does this mean?). If published, this will include your full peer review and any attached files.

Reviewer #1: Yes: Anna Fagre

Reviewer #2: No

Figure Files:

Data Requirements:

Reproducibility:

References

---

## [Editor Report · Decision Letter 1]

29 Jun 2021

Dear Cavalerie,

We are pleased to inform you that your manuscript 'One hundred years of zoonoses research in the Horn of Africa: a scoping review' has been provisionally accepted for publication in PLOS Neglected Tropical Diseases.

Best regards,

Brian Bird, DVM, ScM, PhD

Associate Editor

Stuart Blacksell

Deputy Editor

Dear Authors: Thank you for the submission of your revised manuscript to PLoS NTD. I appreciate your careful consideration and responses to the reviewer's comments and suggestions. I think this work contributes much to our collective view of who is conducting One Health research, where it is happening, and what collaborative teams (global north-south; south-south etc) are writing up the research work from across the region. Yours, -Brian Bird (UC Davis One Health Institute and PLoS NTD Associate Editor)

<style type="text/css">p.p1 {margin: 0.0px 0.0px 0.0px 0.0px; line-height: 16.0px; font: 14.0px Arial; color: #323333; -webkit-text-stroke: #323333}span.s1 {font-kerning: none

</style>

---

## [Editor Report · Acceptance letter]

13 Jul 2021

Dear Cavalerie,

We are delighted to inform you that your manuscript, "One hundred years of zoonoses research in the Horn of Africa: a scoping review," has been formally accepted for publication in PLOS Neglected Tropical Diseases.

Best regards,

Shaden Kamhawi

co-Editor-in-Chief

Paul Brindley

co-Editor-in-Chief
